# Induced Sex Reversal in Adult Males of the Protandric Hermaphrodite *Centropomus undecimalis* Using 17 β-Estradiol: Enhancing Management Strategies for Captive Broodstock

**María de Jesús Contreras-García** [1,2], **Wilfrido Miguel Contreras-Sánchez** [1,*], **Manuel Mendoza-Carranza** [2], **Alejandro Mcdonal-Vera** [1] and **Leonardo Cruz-Rosado** [1]

1. Laboratorio de Acuicultura Tropical, División Académica de Ciencias Biológicas, Universidad Juárez Autónoma de Tabasco, Carretera Villahermosa-Cárdenas Km 0.5, Entronque a Bosques de Saloya, Villahermosa C.P. 86039, Mexico; contrer_mar@hotmail.com (M.d.J.C.-G.); alejandromacdonald_vera@hotmail.com (A.M.-V.); leonardo.cruz@ujat.mx (L.C.-R.)
2. El Colegio de la Frontera Sur. Unidad Villahermosa, Tabasco, Carretera a Reforma Km. 15.5 s/n Ra, Guineo 2da. Sección, Municipio del Centro C.P. 86280, Mexico; xoof1@yahoo.com or mcarranza@ecosur.mx
* Correspondence: contrerw@hotmail.com

**Abstract:** The common snook (*Centropomus undecimalis*) is a protandric hermaphrodite fish that undergoes a sex change during its life cycle. In nature, common snook females develop directly from males shortly after spawning. However, the factors triggering this process remain unknown. This knowledge gap poses challenges for managing the species in captivity. To address this, we conducted a study on sex change induction in three-year-old males using estradiol and evaluated the potential effects of photoperiod manipulation on early maturation. Four treatment groups were employed: (1) fish with estradiol + natural photoperiod; (2) fish without estradiol + natural photoperiod; (3) fish without estradiol + controlled photoperiod; and (4) fish with estradiol + controlled photoperiod. The effectiveness of these treatments was assessed through histological procedures, which allowed for the examination of the fishes' gonads. Furthermore, the concentration of alkali labile phosphorus in fish plasma was measured and correlated with the histological results. Our findings revealed that administering 2 mg/kg estradiol implants resulted in a remarkable 100% female population within the estradiol-treated groups. No significant effect on fish maturation was observed due to the manipulated photoperiod conditions. This protocol offers improved management strategies for captive broodstock. Firstly, the concentration of estradiol used in this study proved sufficient to induce sex change in this hermaphroditic species, enabling the production of viable females at an early age and smaller size and facilitating easier broodstock manipulation. Secondly, the implementation of the alkali labile phosphorus technique allows for sex identification without the need to sacrifice the fish. In conclusion, our study provides valuable insights into sex change induction and photoperiod manipulation in common snook. The findings contribute to enhanced management practices for captive broodstock. However, further research is needed to explore the underlying mechanisms triggering sex change and to optimize protocols for long-term maintenance and successful reproduction in captivity.

**Keywords:** induced sex change; protandric hermaphrodite; common snook; ALP levels; viable females

## 1. Introduction

Fish exhibit a remarkable diversity of reproductive strategies, with hermaphroditic species, including synchronous, protandric (male-first), and protogynous (female-first) hermaphroditism, being relatively common [1–3]. Among fish, teleosts display functional sex reversal as a regular mode of reproduction; this feature has been observed across forty-one families so far [4–6]. In addition, the malleability of the gonads in gonochoric fish

allows for the artificial induction of sex reversal before sexual differentiation occurs, with sex steroids playing a crucial role in directing sexual differentiation [7,8].

Protandric hermaphroditism has been reported in various species of the *Centropomus* genus, including the common snook *Centropomus undecimalis* [9–12], the Mexican snook *C. poeyi* [13], and the fat snook *C. parallelus* [14,15]. Centropomids are euryhaline and diadromous organisms with asynchronous sexual development and fractional spawning [16–19]. Due to their adaptability to diverse environmental conditions, they hold significant potential for aquaculture [20,21]. *C. undecimalis* holds high importance in fishery and commercially along the coasts of Tabasco [22,23] and southern Veracruz [24]. Extensive international research has been conducted on the biology and ecology of this species, aiming to provide insights into conservation and its role as a predator in regulating fish populations [11,17,25–36].

Despite their importance for fishery and ecologically, there is limited information regarding the natural sexual reversion process from male to female; this reproductive feature poses challenges, and the physiological changes that occur during this stage of their life cycle, as well as the mechanisms that induce or regulate this process, are still unknown. However, studies on different Centropomid species, such as *C. poeyi* [13], *C. undecimalis* [35], and *C. parallelus* [36], suggest that exogenous estradiol potentially alters the hypothalamus–pituitary–gonadal mechanism and induces sex change in young males, converting them into functional females [37,38].

In addition, photoperiod manipulation has widely demonstrated its effect on maturation in confined fish, as it can induce oocyte growth, development, and spawning [39–41]. Diverse outcomes have been documented, particularly among species inhabiting temperate climates, including examples such as sea bass (*Dicentrarchus labrax*), zebrafish (*Danio rerio*), and turbot (*Scophthalmus maximus*) [42–44]. Under this mechanism, maturity can be delayed or advanced by the pineal gland, which secretes melatonin, playing a significant role in regulating physiological and behavioral processes during the dark phase of the photoperiod, thereby potentially influencing the timing of maturity in fish. This gland regulates various aspects of physiology and behavior in mammals, including circadian rhythm, sleep, thermoregulation, reproduction, and immunity [45,46].

This study aimed to produce functional females from males through hormonal sex reversal using 17β-estradiol. Additionally, we evaluated the potential acceleration of maturation in newly reversed females through photoperiod manipulation. These findings will contribute to improved handling, reproductive control, and the establishment of strategies for aquaculture management. Furthermore, this study validates using the non-invasive alkali labile phosphorus technique for sex identification without sacrificing the fish, which is highly relevant in daily aquaculture operations.

## 2. Materials and Methods

The study was conducted at the Marine Aquaculture Station facilities (MAS) of the Biological Sciences Academic Division of the Universidad Juárez Autónoma de Tabasco (MAS-DACBiol). All procedures performed followed the ethical standards of the Universidad Juárez Autónoma de Tabasco. When sacrificed for sampling, fish were first killed with an overdose of 1 mL of clove oil per liter of water.

Organisms. Males of three years of age produced in captivity (F1 filial generation) from an induced spawning of wild broodstock were used for the execution of the experiment and physiological monitoring. These fish were maintained until the experiment in 9-m$^3$ geomembrane ponds, with a partial water replacement regime of 90% of the volume every three days. The feed consisted of the commercial marine fish feed Europa® (55% protein, 14% lipids, 8% ash, 0.2% fiber) of different sizes (1–6 mm diameter). Size and length measurements were performed at the experiment's beginning and end to evaluate growth.

Implant elaboration. Implants were composed of cholesterol, cellulose, cocoa butter, and the synthetic steroid 17 β-estradiol (E2; 2 mg/kg of weight); the control implants were

elaborated with all the elements mentioned, except estradiol, following the methodology of Álvarez-Lajonchère and Hernández-Molejón [19].

Experimental design. To induce the production of functional females, a completely randomized factorial design ($2 \times 2$) was used to determine the effects of estradiol implants (with and without E2) and an artificial photoperiod (natural and controlled photoperiod). Before each manipulation, fish were anesthetized using clove oil (0.02 mL per liter of water). Each organism was implanted with a passive transmitter (PIT-TAG) (Avid® Identification Systems, Inc.; Norco, CA, USA) in the dorsal muscle bundle for individual identification. Before applying the cholesterol implants, we verified that the organisms used were all mature males by observing the presence of semen with gentle pressure on the abdomen. Eighty fish were used, distributed in groups of twenty; group one was individually implanted with E2 and kept in natural photoperiod (E2NP); group two was implanted without E2 and maintained in natural photoperiod (0E2NP); group three was implanted without E2 and placed in controlled photoperiod (0E2CP); and group four was implanted with E2 and placed in controlled photoperiod (E2CP). Subsequently, fish were randomly distributed in fiberglass tanks of 7 m³ capacity (3 m Ø, 1 m height) with filtered seawater. Twenty organisms, randomly selected, were placed in each tank (each fish was considered an experimental unit), having a mean weight ($\pm$SD) of 785.25 g ($\pm$62.11) and a total length of 49.61 cm ($\pm$2.18). At the beginning of the experiment, the random distribution generated statistically significant differences between treatments; therefore, initial weight was considered a covariate in the statistical analysis [47]. Feeding was given four times daily (9, 12, 15, and 18 h); the experiment lasted 90 days (11 March–10 June). At the end of the experiment, three fish per treatment were sacrificed for histological analysis to determine sexual maturation progress.

Photoperiod. To determine the effect of the photoperiod on fish maturation, two levels were used (natural photoperiod and controlled photoperiod). In the first group, the natural environmental conditions in the broodstock area of the EAM were maintained during the months of the study. A room with artificial lighting (200 W Volteck® lamps) and lined with black polyethylene was used for the controlled photoperiod. The artificial photoperiod conditions were generated by compressing the natural daylight hours present in the area from March to July. For this purpose, changes in the illumination duration were set at 2.7 min/day. The experiment started with 12 h and 00 min of light (11 March), reaching a maximum daily illumination of 13 h and 20 min (10 April). Then, it was maintained until the last day of the experiment (10 June).

Histological analysis and sex identification. Histological descriptions of the gonads of the sampled fish were performed using the conventional histology technique with hematoxylin–eosin (H–E) staining. To review the complete feminization of the gonad, 6 μm thin sections from the frontal, middle, and caudal portions were taken using a rotary microtome (HM 325; Thermo Scientific Inc., Waltham, MA, USA). Descriptions of ovaries and testes were based on the criteria reported by Grier and Taylor [48] for *C. undecimalis* and by Nakamura et al. [49] and Meijide et al. [50] for other teleost species. The stained sections were examined under a compound microscope (Zeiss®, Oberkochen, Germany), and pictures were taken with an integrated camera ZEISS AxioCam ERc 5s®, Carl Zeiss Microscopy, Thornwood, NY, USA. Fish sampling to verify sexual reversion via cannulation was feasible six months after induction. Spawning induction was conducted during the reproductive season of the following year (12 months after sex reversal).

Detection of alkali labile phosphorus (ALP). To indirectly determine the presence of vitellogenin, the concentration of alkali labile phosphorus was measured following the methodology of Hallgren et al. [51]. Blood samples were taken at 30, 60, and 90 days of experimentation. Blood was drawn from the caudal vein using a fine hypodermic syringe of 3 cc capacity. Subsequently, the samples were transferred on ice to the LAT to separate plasma fluid and serum via centrifugation at $2500\times g$. The samples were mounted on ELISA plates and read in a Stat Fax 303® spectrophotometer Palm City, FL, USA. The results

were expressed in µg $PO_4$/mL plasma. This technique's minimum reliable ALP detection level was 3.2 µg $PO_4$/mL plasma.

Viability of E2-reversed females. One year after the termination of the experiment, three females were selected from both photoperiod treatment groups. The selection process involved cannulation and relied on observations of oocyte retrieval (regardless of their diameter) to select candidates for spawning induction. A 100 µm implant of Ova-RH/kg was inserted into each selected female. Oocyte diameter before implant application, egg diameter, oil drop diameter from spawning females, and fertilization percentage were recorded.

Water quality. Water quality was measured by recording pH and temperature daily with an Ohaus® ST10 potentiometer, dissolved oxygen was measured with a YSI® 5512 multiparameter, and ammonia, nitrites, and nitrates were measured weekly with a Hanna® HI83325 photometer.

Statistical analysis. A two-way ANCOVA was performed to determine the effect of E2 and photoperiod on growth and fish survival, using initial weight as a covariate. The effect of E2 and photoperiod in the percentage of females was analyzed using Fisher's exact test for the factorial contingency table. Plasma ALP levels were determined via simple linear regression. All calibration curves used had high correlation values (r = 0.99). A Kruskal–Wallis test was performed to determine statistical differences in the initial diameter and diameter of fertilized eggs for spawning induction, and a simple ANOVA was performed to identify statistical differences between oil drops of spawned females; all analyses were evaluated according to the postulates of normality and homogeneity of variances [48]. All statistical analyses were carried out using STATGRAPHICS CENTURION® v19.0 software, using a significance level of $\alpha = 0.05$. Graphical analyses were performed using SIGMAPLOT® v14.0 software.

## 3. Results

Sex reversal. Histological analyses of the six fish sampled 90 days after the insertion of implants indicated that fish in treatments receiving E2 (with and without a controlled photoperiod) were 100% female; in contrast, the fish without E2 remained 100% male (Figure 1). In the samples of fish treated with E2, normal ovarian development was observed, the ovarian lumen could be distinguished, and, in the tissue, ovogonia and small germ cells with oval nuclei were visible. Numerous oocytes in the primary growth stage with different sizes were present. Some oocytes were at the multiple nucleoli stage (Figure 1A). In the case of the control groups, testes, defined by the presence of seminiferous tubules in a maturation stage characterized by the presence of spermatogonia, primary spermatocytes, and spermatids, were observed (Figure 1B). At the histological level, no structural differences in gonadal maturation were observed between fish kept under a controlled photoperiod and those held under a natural photoperiod. The average diameter of the oocytes observed histologically indicated an average ($\pm$SD) of $27.11 \pm 10.74$ µm for fish in the natural photoperiod and $28.45 \pm 13.08$ µm for those kept in the controlled photoperiod (ANOVA; *p* > 0.05). At the end of the experiment, seminal fluid was obtained by stripping from all fish not treated with E2 (100% males). Since the E2-treated groups had a reduced diameter gonoduct, introducing the cannula for sampling was impossible. However, cannulation sampling three months later confirmed that 100% of the fish that received E2 were females.

Growth and survival. Fish in the artificial photoperiod treatments had the highest weight due to randomization at the beginning of the experiment; these fish had a slightly higher weight gain (56.75 g) than the natural photoperiod groups (47.58; Table 1). However, the analysis of covariance (ANCOVA) for final weight and total length indicated no significant statistical effects for E2 or photoperiod (*p* > 0.05). Survival was 100% in all treatments.

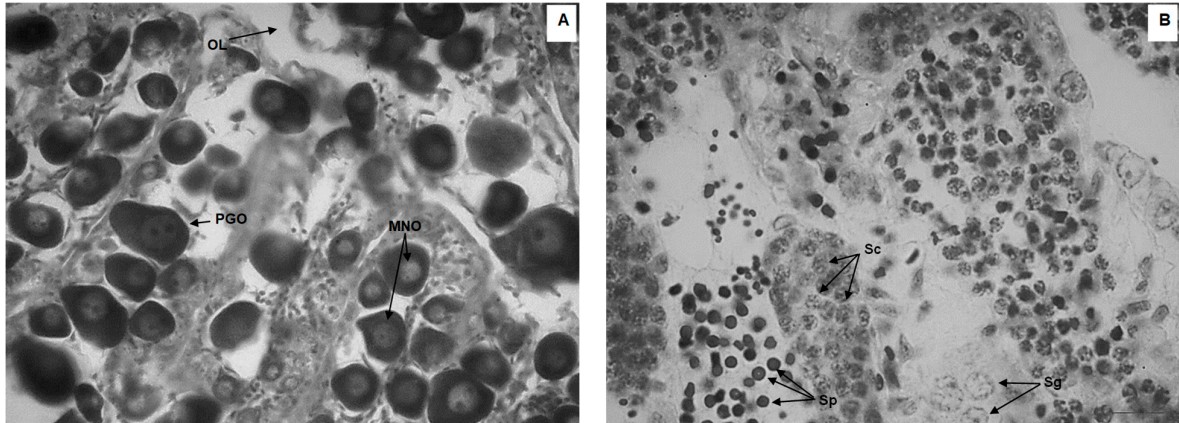

**Figure 1.** Cross section of *C. undecimalis* gonads under 100× objective: (**A**) Ovary from a fish that received an implant with 2 mg of E2 per kg of body weight. (**B**) Testis from a control fish at 100×. OL = ovarian lumen; PGO = oocytes in primary growth stage; MNO = multiple nucleoli oocytes; S g= spermatogonia; Sc = spermatocytes; and Sp = spermatids. Stained with H&E.

**Table 1.** Average values of initial and final weight and total length of *C. undecimalis* used in the experiment.

| Treatment | Initial Weight (g) | Final Weight (g) | Weight Gain (g) | Initial TL (cm) | Final TL (cm) |
|---|---|---|---|---|---|
| E2NP | 768.50 ± 58.45 a | 813.20 ± 62.14 a | 44.70 a | 48.91 ± 2.45 a | 49.60 ± 2.55 a |
| 0E2NP | 765.39 ± 58.98 a | 815.85 ± 63.96 a | 50.45 a | 49.31 ± 2.49 a | 50.94 ± 2.24 a |
| 0E2CP | 807.41 ± 62.69 b | 862.90 ± 87.67 a | 55.50 a | 50.22 ± 1.25 a | 51.57 ± 1.47 a |
| E2CP | 799.70 ± 63.82 b | 857.70 ± 94.08 a | 58.00 a | 49.93 ± 1.65 a | 51.11 ± 1.71 a |

E2NP = estradiol + natural photoperiod; 0E2NP = control + natural photoperiod; 0E2CP = control + controlled photoperiod; and E2CP = estradiol + controlled photoperiod. N = 20 in each treatment. Different letters in a column indicate statistically significant differences between treatments.

Alkali labile phosphorus (ALP). Statistical analysis indicated a highly significant statistical effect on ALP levels related to using E2 (ANOVA; $p < 0.001$). However, no photoperiod, time, or factor interaction effects were observed ($p > 0.05$). Throughout the experiment, fish with estradiol averaged 28.62 ± 18.11 μg/mL of ALP. At 30 days, fish with E2 averaged 35.42 ± 13.27 μg/mL; at day 60, the fish averaged 17.21 ± 19.54; and by day 90, the mean was 33.24 ± 18.76 μg/mL. Fish without E2 always presented values below the detection limits (Figure 2A–C).

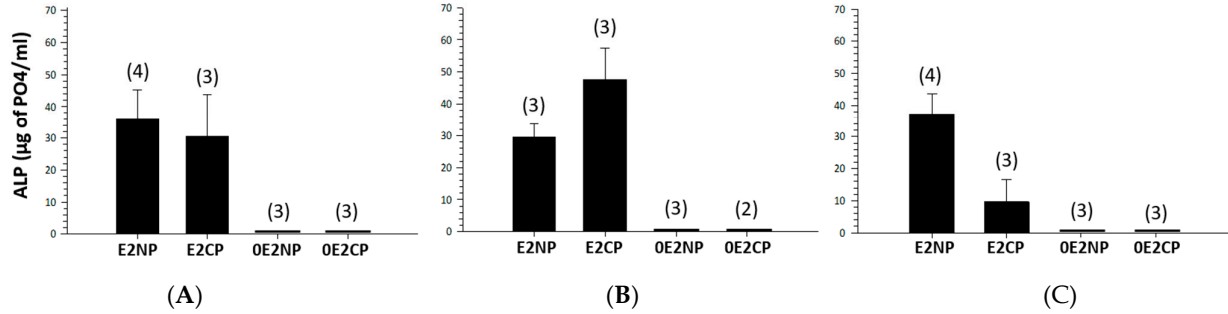

**Figure 2.** Alkali labile phosphorus (ALP) concentrations in *C. undecimalis* at 30 days (**A**), 60 days (**B**), and 90 days (**C**) of experimentation. Numbers in parentheses indicate the sample size per treatment. E2NP = estradiol + natural photoperiod; 0E2NP = control + natural photoperiod; 0E2CP = control + controlled photoperiod; and E2CP = estradiol + controlled photoperiod.

Viability of reversed females. Twelve months after sexual reversion, all females had growing oocytes. The average diameter of the oocytes obtained via cannulation was 79.76 ± 14.46 μm for females that were part of the natural photoperiod group and

71.96± 15.40 μm for those under the controlled photoperiod. The two groups had no statistical differences (KW; $p > 0.05$). Two females from the natural photoperiod and two from the controlled photoperiod treatments spawned from the six females induced (66.7%). The diameter of the eggs obtained from the spawnings showed no statistically significant differences (KW; $p > 0.05$), averaging 685.37 ± 30.74 μm (Table 2). No significant statistical differences were observed regarding the oil drop diameter present in the oocytes, averaging 176.43 ± 12.41 μm (ANOVA; $p > 0.05$).

**Table 2.** Egg and oocyte measurements as a reference to the viability of females reverted with E2 that were maintained in natural and controlled photoperiods.

| Treatment | Initial Ø (μm) * | Final Ø (μm) ‡ | Spawning | Egg Ø (μm) | Oil Drop Ø (μm) |
|---|---|---|---|---|---|
| E2CP | 96.22 ± 100.30 | n/a | Yes | 681.13 ± 19.91 | 177.78 ± 11.12 |
| E2CP | 73.97 ± 75.73 | n/a | Yes | 687.11 ± 19.91 | 179.47 ± 12.13 |
| E2CP | 69.10 ± 59.65 | 44.30 ± 13.97 | No | n/a | n/a |
| E2NP | 89.72 ± 83.20 | n/a | Yes | 687.74 ± 49.47 | 172.69 ± 12.41 |
| E2NP | 62.34 ± 15.92 | n/a | Yes | 685.48 ± 22.86 | 175.78 ± 13.97 |
| E2NP | 63.81 ± 14.47 | 49.61 ± 12.17 | No | n/a | n/a |

E2NP = estradiol + natural photoperiod; E2CP = estradiol + controlled photoperiod; n/a = no data. Egg measurements were obtained from the four females that spawned. Two females under the same maturation status did not spawn; therefore, samplings were conducted via cannulation to observe oocyte condition. No statistically significant differences between photoperiod treatments were observed. Ø = diameter; * before spawning induction; ‡ after induction and no spawning.

Water quality. The water quality during the experiment remained within levels suitable for marine fish culture. No statistical differences were observed between treatments for any of the registered parameters. However, the temperatures in the controlled photoperiod were, on average, 1.44 degrees higher than those registered in the natural photoperiod (27.72 vs. 26.28 °C). Dissolved oxygen maintained an average of 4.31 ± 1.17, and mean salinity was 32.01 ± 0.34 UPS in all tanks. Ammonium concentrations were, on average, 0.47 ± 0.38 mg/L, nitrate 2.13 ± 1.65 mg/L, and nitrite 0.18 ± 0.24 mg/L (Table 3).

**Table 3.** Average water quality values in the tanks of each treatment.

| Treatment | DO (mg/L) | Temperature (°C) | pH (UI) | Ammonia (mg/L) | Nitrates (mg/L) | Nitrites (mg/L) |
|---|---|---|---|---|---|---|
| E2NP | 4.38 ± 1.15 | 26.26 ± 1.87 | 8.56 ± 0.63 | 0.21 ± 0.22 | 1.86 ± 1.38 | 0.11 ± 0.13 |
| 0E2NP | 4.39 ± 1.20 | 26.29 ± 1.91 | 8.52 ± 0.66 | 0.53 ± 0.39 | 1.84 ± 1.39 | 0.08 ± 0.11 |
| 0E2CP | 4.20 ± 1.18 | 27.75 ± 1.65 | 8.79 ± 2.31 | 0.60 ± 0.44 | 2.22 ± 1.92 | 0.21 ± 0.20 |
| E2CP | 4.25 ± 1.16 | 27.70 ± 1.66 | 8.48 ± 0.73 | 0.52 ± 0.40 | 2.30 ± 2.14 | 0.27 ± 0.40 |

E2NP = estradiol + natural photoperiod; E2FC = estradiol + controlled photoperiod. No statistical differences were found among treatments.

## 4. Discussion

This research demonstrates the efficacy of E2 in 2 mg/kg implants, allowing for the early production of *C. undecimalis* females from captive-produced three-year-old males. Although three species of the genus (*C. poeyi*, *C. undecimalis*, and *C. parallelus*) have been successfully reverted to date, in most cases, sexual reversion has been performed on juveniles [36,37,52], and although Passini et al. [33] reported successful reversion in juvenile males of *C. undecimalis*, the viability of the females obtained with the doses used in those studies was not demonstrated. In our laboratory, this is the second Centropomid species in which viable females have been obtained from young mature males, as we previously succeeded in reproducing sex-reversed females of *C. poeyi* [13]. The cultivation of this species benefits from the production of smaller females due to its protandric hermaphroditic nature. Most females only appear several years later when the fish reach a weight of 3–5 kg.

Consequently, these smaller females require less space and consume lower quantities of food, making it easier to manage the broodstock. This management strategy enables an average production of half a million eggs per female during spawning.

We initiated the present study with mature males showing spermatozoa available for reproduction. Histologically, no evidence of primary females was observed, as [53] reported in the protandric hermaphrodite barramundi (*Lates calcarifer*). As a result of the E2 implantation, only female tissue was found in the gonads, showing developing lamellae and oocytes in the initial stage of development, ovogonia with oval nucleoli, and oocytes in primary growth, demonstrating the total suppression of testicular tissue. Our findings are similar to those presented by Vidal-López et al. [13] in *C. poeyi*, De Carvalho et al. [36] in *C. parallelus*, and Passini et al. [33] and De Carvalho et al. [36] in *C. undecimalis*. It has been reported that administering steroids for sex change during sexual differentiation may affect the responsiveness of developing germ and somatic cells by disrupting the gonadal pathway [54]. This interference with sex change is due to exogenous steroids functioning as endocrine disruptors [55], and even remnants of natural and synthetic estrogens affect the endocrine and reproductive systems and contribute to the development of secondary male and female characteristics, consequently diverting or reprogramming sexual differentiation [56].

Devlin and Nagahama [7] and Guiguen et al. [57] have documented that sex-specific gene networks maintain sexual fate in fish, promoting a steroidogenic environment according to the expressed sex. However, administering exogenous steroids at the precise time leads to sex reversal, altering hormone secretion in fish. Devlin and Nagahama [7] mention that gonadogenesis involves multilevel cellular control communication in a complex manner involving biochemical, neurological, and physiological pathways to provide the necessary plasticity for gonadal development to proceed in context with intrinsic and environmental factors. Authors such as Nagahama [58] and Devlin and Nagahama [7] suggest that the balance of steroid hormones in sex-changing fish is ultimately regulated via the hypothalamus–pituitary–gonadal (HPG) axis. In this regard, it has been proposed that gonadotropin-releasing hormone (GnRH), generated in the hypothalamus, stimulates the pituitary to produce and release follicle-stimulating gonadotropin hormones (FSH) and luteinizing hormone (LH) into the bloodstream. These gonadotropic hormones are primarily responsible for regulating steroidogenesis in the gonads [59]. The factor that triggers sex reversal in fish that first mature as males before becoming females remains unclear. It has been proposed that a threshold age or size may trigger sex inversion in protandrous species. However, Guiguen et al. [60] consider that this condition may not apply to protandric species that spawn in groups. We speculate that these protandrous species will live as males until they encounter an environmental stressor that triggers sex change. In the case of protogynous fish, Liu et al. [61] proposed that sex change involves a perturbation in the HPG axis caused by the hypothalamic–pituitary–interrenal (HPI) axis, with cortisol acting as a key factor. Cortisol may block *Cyp19a1a* transcription, triggering a chain reaction that promotes a decrease in estradiol levels, accelerating ovarian degeneration and disrupting female-specific gene expression. Despite little evidence, Todd et al. [38] proposed that the redirection of gonadal fate initiates when the normal expression of key genes responsible for maintaining sex identity is disrupted (*cyp19a1a* in protogynous species and *dmrt1* in protandrous species). This disruption leads to a sequential breakdown of the existing expression patterns, hormonal balance, and anatomical structure within the gonads. Once the inhibitory influence on the opposite sexual network is removed, a new sex-specific gene expression and hormonal environment takes over, guiding the development of the gonads towards secondary sexual characteristics. This evidence leads us to propose that the application of exogenous E2 acts at the second phase of the model proposed by Todd et al. [38], altering the natural process by saturating the HPG feedback system and activating the genetic and physiological events required to maintain a female steroidogenic environment. The sustainment of gonadal change and viability of females produced in this study and *C. poeyi* indicate that this process is irreversible [13].

In natural populations, Taylor et al. [10] found that female common snook develop directly from previously spawned males, that sex transition occurs shortly after spawning, and that they can spawn as females during the following breeding season. This finding has been verified in our study, as females immediately after the sex change were maturing. These newly reverted fish need a morphological reorganization of the gonadal ducts to allow the expulsion of eggs, making them able to spawn as females in the next annual reproductive event.

The detection of ALP as a sex indicator in sex-reversed organisms of *C. undecimalis* represents a significant result in this study. The data indicate that females receiving E2 had reliable values above the detection limit three months after implantation. At the same time, males in the control groups always remained at values below this limit. According to Craik and Harvey [62], concentrations of 10 µg/mL of ALP are sufficient to assume the existence of a mature or maturing female. Therefore, it is possible to affirm that sexual reversion in fish had already been consummated by this period, and the ovaries were in total development. Hernández-Vidal et al. [63] indicated that the concentration of ALP in *C. undecimalis* females from marine and freshwater environments had two periods of increase (from April to May and from August to September), reaching the maximum value in the second period with 83 µg/mL. In the freshwater environment, females of the same reproductive stage and with vitellogenic oocytes had levels comparable to those observed in the marine environment. In contrast, the lowest ALP levels were observed in females with previtellogenic oocytes. The results of these authors coincide with those found in our study for the sex-reversed females with ovaries under early development. In addition, the levels observed are between values reported for females of other species in the wild and captive species induced with E2. Such is the case reported by Emmersen and Petersen [64], who determined that E2 increases ALP levels in both sexes of sole (*Platichtys flesus*). They reported that normal males and non-vitellogenic females had values just above 4.0 µg/mL, while treated males averaged 65.36 µg/mL and females in vitellogenesis had 86.30 µg/mL. Whitehead et al. [65] observed basal levels of ALP in rainbow trout, very similar to those reported for sole. However, towards the reproductive period, females reached levels of up to 400 µg/mL, with no changes observed in males. Nagler et al. [66] mentioned that in trout, there is a direct relationship between the level of vitellogenin in blood and some indirect indicators such as ALP, making it possible to distinguish vitellogenic fish from non-vitellogenic fish and, therefore, to determine the reproductive status of the fish. This idea was reinforced by Waagboe and Sandnes [67], who identified that in rainbow trout, there is a robust correlation between ALP levels and VTG, so it is considered an excellent method to detect sexual maturation in this species.

Our results indicated no significant structural differences in the gonadal development of females maintained under controlled or natural photoperiods. This suggests that photoperiod does not alter the process or accelerate the timing in these fish under induced transition from male to female. The effect of photoperiod on gonadogenesis has been widely documented. Annual changes in photoperiod act on the pineal gland and hypothalamus, which secrete and synthesize hormones responsible for reproduction, such as gonadotropin-releasing hormones, regulating, in turn, the production of gonadal steroids [68–70]. Although its role as an environmental signal in tropical fish reproduction is under-represented [71], its effect is undeniable. Our study suggests that, at least during early ovarian differentiation in sex-reversed females, the effects are not significant.

Important issues in using synthetic steroids, in addition to sex change, are achieving good growth and survival in the reverted fish. We found no statistical effects of E2 on the growth of experimental fish. According to Pandian and Sheela [72], there needs to be more consistent data on this issue since some studies indicate anabolic effects while others found no or adverse effects. The literature contains multiple examples of contradictory information [34,73,74]. In Centropomids, several studies reflect these contradictory findings; for example, De Carvalho et al. [37] found that common snook treated with E2 had lower growth regardless of the dose when compared with the control group, reporting at least a

7 g difference. Similarly, Banh et al. [54] indicated that when using E2, the fish grew less than the control group in Asian snook. In contrast, Vidal-López et al. [53] found the best growth for common snook when treated with E2 for 21 days.

In the present investigation, the treatment with E2 did not affect the survival of the fish; therefore, the use of implants to administer the steroid is recommended, avoiding adverse effects on the organisms and obtaining 100% survival. Similar results have been observed in other studies where E2 has been used for sex change in common snook [53]; false clownfish [75]; Mexican snook [13]; and Asian snook [54]. However, there is contradictory information where it is evident that using E2 can cause severe mortality; according to Pandian and Sheela [72], treatment with a synthetic steroid generally causes high mortality in most species. These contradictory results might be associated with the dose used, as Passini et al. [34] reported, since they achieved 100% survival in common snook treated with E2 at 0.5 and 1.0 mg/kg. In contrast, at quantities of 8 mg/kg, fish did not survive.

The viability of female Centropomids reversed with E2 has been demonstrated with the results of this work and those of Vidal-López et al. [13] with Mexican snook. In both cases, it was possible to obtain successful spawning during the natural spawning season, even when the fish had a small oocyte diameter, as indicated in previous studies [34,76–78]. We observed 100% of the females maturing and 66.7% of females spawned in each group treated under controlled and natural photoperiod conditions one year earlier. These values can be improved by inducing ovarian maturation and synchronization. We consider that, in effect, the administration of E2 via implants represents an essential advantage for all the tissues to adjust, allowing an efficient sex change and subsequent spawning.

The effectiveness of steroid administration relies on various factors, including the specific type of steroid, water temperature, age, and other relevant considerations, depending on the treated species [79]. Induced sex reversal is most successful when the effective treatment (type and hormone dose) is administered when the gonads are most sensitive to exogenous hormones (labile period). The fact that *C. undecimalis* is a protandric hermaphrodite and possesses bipotential gonads during its male stage makes it permanently susceptible to initiating sex change in response to chemical or physical stimuli. This allows the exogenous steroid to exert effects on the sex change mechanism at any time in the life history of this species, as long as it is male. Based on this, we assert that the fish used in this study were within a period where they were sensitive to E2 exposure and that adequate methods and doses were used.

## 5. Conclusions

Viable females of *C. undecimalis* were obtained in captivity using estradiol. A successful implantation protocol was generated, with adequate hormone concentration, avoiding using food as a vehicle for estradiol, which prevents manipulation of the steroid and its potential release into the environment. Using ALP as a sex identifier is beneficial in managing common snook in captivity. It makes it possible to differentiate females from males with a minimally invasive technique that does not involve the damage to gonadal tissues that cannulation implies.

**Author Contributions:** All authors contributed to the study conception and design, material preparation, and data collection. M.d.J.C.-G. and W.M.C.-S. performed data analysis. The first draft of the manuscript was written by M.d.J.C.-G. and W.M.C.-S., and all authors commented on previous versions of the manuscript. All authors have read and agreed to the published version of the manuscript.

**Funding:** This work was supported by Universidad Juárez Autónoma de Tabasco. (DV/DGPYS/017). María J. Contreras-García received research support under the scholarship program UJAT-PISA. We thank the Consejo Nacional de Humanidades, Ciencias y Tecnologías (CONAHCYT) and El Colegio de la Frontera Sur for tuition exemption.

**Institutional Review Board Statement:** All procedures performed were following the ethical standards of the Universidad Juárez Autónoma de Tabasco. When sacrificed for sampling, fish were first anesthetized with an overdose of clove oil (1 mL/L of water).

**Informed Consent Statement:** Not applicable.

**Data Availability Statement:** Data supporting this study will be available from UJAT's repository.

**Acknowledgments:** We extend our sincere appreciation to the dedicated personnel at the marine station whose unwavering commitment and expertise were instrumental in the successful execution of the experiments presented in this paper. Their meticulous care and attention to detail ensured the well-being and optimal conditions for the fish under study, thereby contributing to the robustness and reliability of our findings.

**Conflicts of Interest:** The authors declare no conflict of interest.

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
