# Peer review of "Induced Sex Reversal in Adult Males of the Protandric Hermaphrodite Centropomus undecimalis Using 17 β-Estradiol: Enhancing Management Strategies for Captive Broodstock"

_2673-9496, doi:10.3390/aquacj3030016_

Round 1

Reviewer 1 Report

In this study, the authors conducted a study on sex change induction in male common snook using estradiol and evaluated photoperiod manipulation's potential effects on early maturation. This work fits the scope of the Aquaculture Journal and make a contribution to some new knowledge in management practices for captive broodstock. However, I have some comments and there still exists room to improve.

1 As the authors mention, "the exact developmental stage at which this transition occurs and the factors influencing it remain unknown", what are the developmental characteristics of gonads during sex reversal? A series of histological analyses at the different time points of this process may be beneficial for understanding the effects of E2 to sexual reversal.

2 There are some elements missing in Figure 2 (e.g. A/B/C, numbers in parentheses). The time points before 30 days may provide some important information about sex reversal and vitellogenesis. Furthermore, I suggest the authors change this figure to make the results more intuitive.

3 In the study, they found no significant structural differences in the gonadal development, weight and total length of females maintained under controlled or natural photoperiods. I doubt if the experimental conditions for control photoperiods were not set appropriately.

4 The results of Table 2 are confusing, especially for the readers not in this field. Initial oocyte prior to spawning induction(stage IV) and egg (stage V), as well as the final oocyte in non-spawning females (why use these two abnormal fish) should be similar in size. Why are they so different? 

5 Some scientific names are not italic. e.g. C. undecimalis at lines 192, 203, and 226.

Author Response

In this study, the authors conducted a study on sex change induction in male common snook using estradiol and evaluated photoperiod manipulation's potential effects on early maturation. This work fits the scope of the Aquaculture Journal and make a contribution to some new knowledge in management practices for captive broodstock. However, I have some comments and there still exists room to improve.

1 As the authors mention, "the exact developmental stage at which this transition occurs and the factors influencing it remain unknown" what are the developmental characteristics of gonads during sex reversal? A series of histological analyses at the different time points of this process may be beneficial for understanding the effects of E2 to sexual reversal.

Since the sentence refers to individuals, not to gonads, it was modified for clarity as follows in lines 13-14.

“In nature, common snook females develop directly from males shortly after spawning. However, the factors triggering this process remain unknown”.

The reviewer is right regarding the close analysis of morphological changes at different times. The idea of analyzing histologically gonads that were induced to change from testis to ovaries is currently being assessed in a new experiment. It is important to mention that the present study focused on the feasibility of sex change and the viability of females.

2 There are some elements missing in Figure 2 (e.g. A/B/C, numbers in parentheses). The time points before 30 days may provide some important information about sex reversal and vitellogenesis. Furthermore, I suggest the authors change this figure to make the results more intuitive.

We appreciate the comments, some features in this figure were not displayed. We have modified the figure for clarity, and the missing items were placed.

3 In the study, they found no significant structural differences in the gonadal development, weight and total length of females maintained under controlled or natural photoperiods. I doubt if the experimental conditions for control photoperiods were not set appropriately.

We guarantee that the setting for the photoperiod was adequate. The plastic cover generated a dark room and artificial illumination provided the specific hours of light. As we discussed, this result indicates that the photoperiod protocol implemented in our study did not influence early gonadal development. We have used the same setting for final maturation in gray snapper, significantly affecting the number of fish ripe and sperm quantity and quality.

4 The results of Table 2 are confusing, especially for the readers not in this field. Initial oocyte prior to spawning induction(stage IV) and egg (stage V), as well as the final oocyte in non-spawning females (why use these two abnormal fish) should be similar in size. Why are they so different? 

We have modified the table to make it clear to the reader.

5 Some scientific names are not italic. e.g. C. undecimalis at lines 192, 203, and 226.

We have italicized the scientific names.

Reviewer 2 Report

The manuscript ID: aquacj-2543542 - " Induced sex reversal in adult males of the protandric hermaphrodite Centropomus undecimalis, using 17 β-Estradiol: Enhancing Management Strategies for Captive Broodstock" provides an up-to-date and important information on the use of estrogens in inducing sex reversal in adult males of the protandric hermaphrodite fishes. This data is relevant for a broad public since this knowledge gap poses challenges for managing the species in captivity. It is perfectly clear to anyone familiar with this kind of study that it had to be a great effort for authors to plan and conduct such a study and I have a great appreciation for this. This study provides valuable insights into sex change induction and photoperiod manipulation in common snook.

In my opinion, the writing manner is very effective in uptake of prominent and key points of each research work, the way presentation of the manuscript is very good, however grammatical and technical error issue arises and it may affect the manuscript quality. Thus, the authors need to pay much attention to eliminate this kind of errors to provide a good quality to the paper for publication. The background provides sufficient literature, the material is explanatory and well-discussed. Overall, a good read.

I support the publication of this paper in Aquculture Journal following a minor revision.

Grammatical and technical error issue arises and it may affect the manuscript quality, thus, the authors need to pay much attention to eliminate this kind of errors to provide a good quality to the paper for publication. 

Author Response

In my opinion, the writing manner is very effective in uptake of prominent and key points of each research work, the way presentation of the manuscript is very good, however grammatical and technical error issue arises and it may affect the manuscript quality. Thus, the authors need to pay much attention to eliminate this kind of errors to provide a good quality to the paper for publication. The background provides sufficient literature, the material is explanatory and well-discussed. Overall, a good read.

A thorough review of the entire paper was conducted. Grammatical and technical errors were corrected.

I support the publication of this paper in Aquaculture Journal following a minor revision.

We truly appreciate this comment.

Comments on the Quality of English Language. Grammatical and technical error issue arises and it may affect the manuscript quality, thus, the authors need to pay much attention to eliminate this kind of errors to provide a good quality to the paper for publication. 

A thorough review of the entire paper was conducted. Grammatical and technical errors were corrected.

Reviewer 3 Report

The authors investigated the sex-changing mechanism of the common snook (Centropomus undecimalis) by treating three-year-old males with or without estradiol and photoperiod manipulations. They assessed the effects on gonadal histology and the concentration of alkali labile phosphorus in fish plasma. They found that estradiol administration resulted in all female population. However, manipulation of photoperiod did not induce significant effects on fish maturation. They concluded that this protocol offers improved management strategies for captive brood stock, enabling the production of viable females at an early age with smaller size and facilitating easier brood stock manipulation. In addition, the implementation of the alkali labile phosphorus technique allows for sex identification without the need to sacrifice the fish. Their study provides valuable insights into sex change induction and photoperiod manipulation in common snook and contributes to enhanced management practices for captive brood stock. However, further research is needed to explore the underlying mechanisms triggering sex change and to optimize protocols for long-term maintenance and successful reproduction of common snook in captivity.

General Comments:

The manuscript is written clearly and concisely. The methods and results are sound.

Specific Comments:

Line 61: hypothalamus-pituitary-gonadal…

Line 134: Delete “5”.

Lines 226-229: Where are the “numbers in parentheses” in Figure 2? The error bars are not complete on my computer screen.

Line 297: hypothalamus-pituitary-gonadal…

Lines 309-312: When cyp19a1a is blocked, shouldn’t testosterone level be higher since it will not be converted into estradiol?

The quality of English Language is fine.

Author Response

General Comments:

The manuscript is written clearly and concisely. The methods and results are sound.

We truly appreciate the comments on our paper

Specific Comments:

Line 61: hypothalamus-pituitary-gonadal…

This sentence was corrected.

Line 134: Delete “5”.

The number was deleted.

Lines 226-229: Where are the “numbers in parentheses” in Figure 2? The error bars are not complete on my computer screen.

We appreciate the observations, during the inclusion of the figure, some features were not displayed. We have modified the figure for clarity, and the missing items were placed.

Line 297: hypothalamus-pituitary-gonadal…

This sentence was corrected.

Lines 309-312: When cyp19a1a is blocked, shouldn’t testosterone level be higher since it will not be converted into estradiol?

We appreciate this observation, we mistakenly described the process for protogynous fish as an example. This was corrected in the final version as follows:

“In the case of protogynous fish, Liu et al [62] proposed that sex change involves a perturbation in the HPG axis caused by the hypothalamic-pituitary-interrenal (HPI) axis, with cortisol acting as a key factor. Cortisol may potentially block Cyp19a1a transcription, thereby triggering a chain reaction that promotes a decrease in estradiol levels, accelerating ovarian degeneration and disrupting female-specific gene expression.

Comments on the Quality of English Language

The quality of English Language is fine.

A thorough review of the entire paper was conducted. Grammatical and technical errors were corrected.

Round 2

Reviewer 1 Report

I have no comments.